# Training-free Long Video Generation with Chain of Diffusion Model Experts

## Abstract

Video generation models hold substantial potential in areas such as filmmaking. However, current video diffusion models need high computational costs and produce suboptimal results due to high complexity of video generation task. In this paper, we propose **ConFiner**, an efficient high-quality video generation framework that decouples video generation into easier subtasks: structure **con**trol and spatial-temporal re**fine**ment. It can generate high-quality videos with chain of off-the-shelf diffusion model experts, each expert responsible for a decoupled subtask. During the refinement, we introduce coordinated denoising, which can merge multiple diffusion experts' capabilities into a single sampling. Furthermore, we design ConFiner-Long framework, which can generate long coherent video with three constraint strategies on ConFiner. Experimental results indicate that with only 10% of the inference cost, our ConFiner surpasses representative models like Lavie and Modelscope across all objective and subjective metrics. And ConFiner-Long can generate high-quality and coherent videos with up to 600 frames. All the code will be available at project website: `https://confiner2025.github.io`.

## 1 Introduction

Generative AI Cao et al. (2023); Zhang et al. (2023a); Wu et al. (2023) has recently emerged as a hotspot in research, influencing various aspects of our daily life. For visual AIGC, numerous image generation models, such as Stable Diffusion Rombach et al. (2022) and Imagen Saharia et al. (2022), have achieved significant success. These models can create high-resolution images that are rich in creativity and imagination, rivaling those created by human artists. Compared to image generation, video generation models Ho et al. (2022c;b); Xing et al. (2023); Esser et al. (2023) hold higher practical value with the potential to reduce expenses in the fields of filmmaking and animation.

However, current video generation models are still in their early stages of development. Existing video diffusion models can primarily be categorized into three types. The first type uses T2I (Text to Image) models to generate videos directly without further training Khachatryan et al. (2023). The second type incorporates a temporal module into T2I models and trains on video datasets Wang et al. (2023b); Blattmann et al. (2023); Lin & Yang (2024). The third type is trained from scratch Ma et al. (2024). Regardless of which type, these methods use a single model to undertake the entire task of video generation. However, video generation is an extremely intricate task. After our in-depth analysis, we believe that this complex task consists of three subtasks: modeling the *video structure*, which includes designing the overall visual structure and plot of the video; generating *spatial details*, ensuring that each frame is produced with sufficient clarity and high aesthetic score; and producing *temporal details*, maintaining consistency and coherence between frames to ensure natural and logical transitions. Therefore, relying on a single model to handle such a complex and multidimensional task is challenging.

Overall, there are three main challenges in the current field of video generation. 1) The quality of the generated videos is low, hard to achieve high-quality temporal and spatial modeling simultaneously. 2) The generation process is time-consuming, often requiring hundreds of inference steps. Utilizing a single model to handle complex video generation task is one of the key reasons for these two issues. 3) The length of the generated videos are typically short. Due to limitations in VRAM, the length of videos generated in a single attempt generally ranges between only 2-3 seconds.

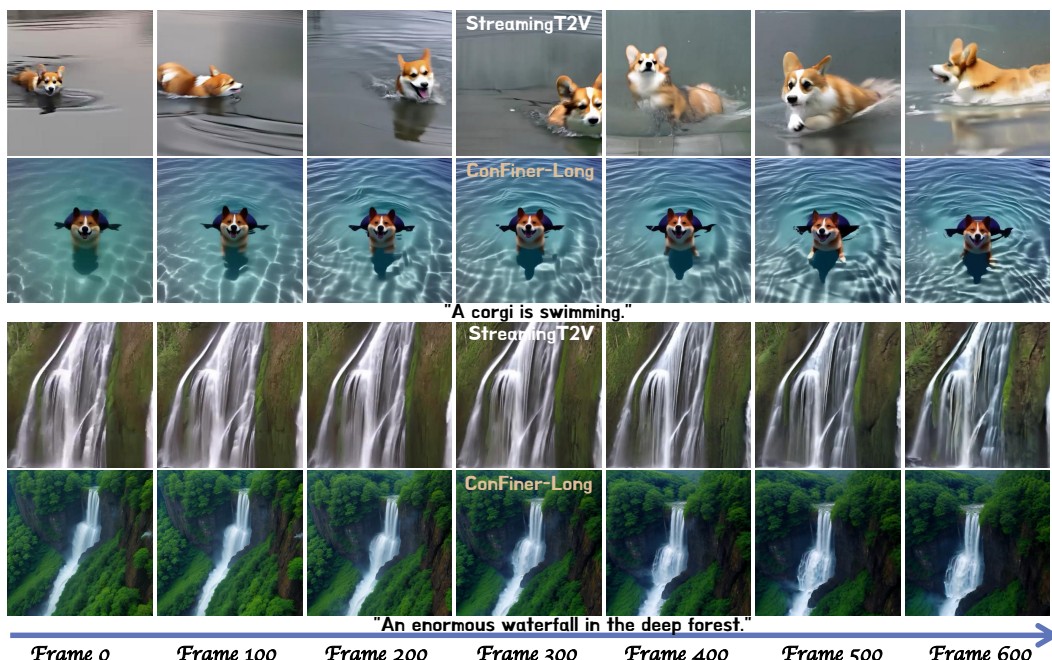

Figure 1: Comparison between Our ConFiner-Long and StreamingT2V Henschel et al. (2024).

In order to enhance generation quality, some methods employ multiple models on different resolutions or in different spaces to perform progressive generation. Methods like Imagen Video Ho et al. (2022a) and I2VGen-XL Zhang et al. (2023c) train several diffusion models on gradually increasing resolutions to first generate a video of low resolution, and then progressively scale up. Show-1 Zhang et al. (2023b) trains a model in pixel space to generate low-quality videos, followed by a latent space model to enhance quality. Compared to methods using a single model, these approaches achieve higher performance. However, each model still needs to handle both spatial and temporal modeling in generation. This leaves each model still heavily burdened.

To improve quality of videos while reducing inference time, we rethink the demands of video generation tasks, which include modeling video structure, generating spatial details, and producing temporal details. We find out that a more rational approach is utilizing three specialized models, each handling one demand. By doing so, these models can collaboratively accomplish the comprehensive task of video generation. To this end, we propose a framework named ConFiner, which decouples the video generation process into three parts: structure control, temporal refinement, and spatial refinement. During generation, we employ chain of three ready-made diffusion experts, each specializing in their respective tasks. In the control stage, a highly controllable T2V (Text to Video) model is employed as control expert, tasked with structure control. During the refinement stage, a T2I model and a T2V model skilled at generating details are employed as spatial and temporal experts to refine details. This framework can reduce the burden on individual models, enhancing both the quality and speed of generation. Moreover, as it utilizes ready-made diffusion experts, this framework does not incur additional training costs.

Moreover, during refinement stage, to enable simultaneous use of spatial and temporal experts within a single denoising process, we proposed coordinated denoising. Due to the varying noise schedulers employed by different diffusion experts, their noise distributions differ. Therefore, they cannot be directly combined during generation. In coordinated denoising, we build a bridge between two noise schedulers, enabling the collaboration of two diffusion models at the granularity of timesteps.

In terms of increasing video lengths, some methodsWang et al. (2024; 2023a); Qiu et al. (2023a); Gu et al. (2023) propose generating video segments and then stitching them together to create longer videos. They use techniques such as noise control to ensure consistency between segments. Although these methods can produce extended videos within limited VRAM, the transitions between segments tend to be abrupt.

Therefore, based on ConFiner, we propose ConFiner-Long framework with three strategies to ensure the coherence and consistency between segments in long video generation. As the initial noise significantly impacts the final videos, we first introduce a segments consistency initialization strategy to ensure the consistency of the initial noise between segments by sharing a base noise. Additionally, in order to enhance the coherence of the motion modes between segments, we propose a coherence guidance strategy that uses the gradient of noise differences between two segments to guide the denoising direction. Also, to address the problem of flickering at the junctions of segments, we design a staggered refinement strategy that staggers the control stage and the refinement stage. It places the tail of one video structure and the head of the next into the same refinement process to achieve more natural transitions between segments.

Experimental results have shown that ConFiner requires only 9 sampling steps (less than 5 seconds) to surpass the performance of models like AnimateDiff-Lightning Lin & Yang (2024), LaVie Wang et al. (2023c), and ModelScope T2V Wang et al. (2023b) with 100-step sampling (more than 1 minute) in all metrics. Furthermore, ConFiner-Long can generate high-quality coherent videos up to 600 frames long. To sum up, our contributions are as follows:

1. We introduced ConFiner, which decouples the video generation task into three sub-tasks. It utilizes three ready-made diffusion experts, each handling its specialized task. This approach reduces the model's burden, enhancing the quality and speed of generation.

2. We developed coordinated denoising, allowing two experts on different noise schedulers to collaborate timestep-wise in video generation process.

3. We proposed ConFiner-Long framework, building on the ConFiner with three strategies to achieve high-quality, coherent long video production.

## 2 RELATED WORK

**Diffusion models (DMs).** DMs have achieved remarkable successes in the generation of images Nichol & Dhariwal (2021), music Mittal et al. (2021), and 3D models Poole et al. (2022); Lin et al. (2023). These models typically involve thousands of timesteps, controlled by a scheduler that manages the noise level at each step. Diffusion models consist of two processes. In the forward process, noise is progressively added to the original data until it is completely transformed into noise. During the reverse denoising process, the model starts with random noise and gradually eliminates the noise using a denoising model, ultimately transforming it into a target sample.

**Video Diffusion Models (VDMs).** Compared to the success of diffusion models in image generation and other areas, VDMs are still at a very early stage. Models like text2video-zero Khachatryan et al. (2023) that use stable diffusion without additional training for direct video generation suffer from poor coherence and evident visual tearing. Models like SVD Blattmann et al. (2023) and Modelscope T2V Wang et al. (2023b) convert the U-Net of stable diffusion Wang et al. (2023c) into a 3D U-Net through the addition of temporal convolution or attention, and train it on video datasets to achieve video generation. Although these video generation models each have their strengths, none fully satisfy all the demands of video generation, such as coherence and clarity.

## 3 METHOD

### 3.1 OVERVIEW

Our ConFiner consists of two stages: the control stage and the refinement stage. In the control stage, it generates a video structure containing coarse-grained spatio-temporal information, which determines the overall structure and plot of the final video. During the refinement stage, it refines spatial and temporal details based on video structure. In this stage, we propose coordinated denoising to enable cooperation of spatial expert and temporal expert. Based on ConFiner, we introduce ConFiner-Long framework for producing coherent and consistent long videos.

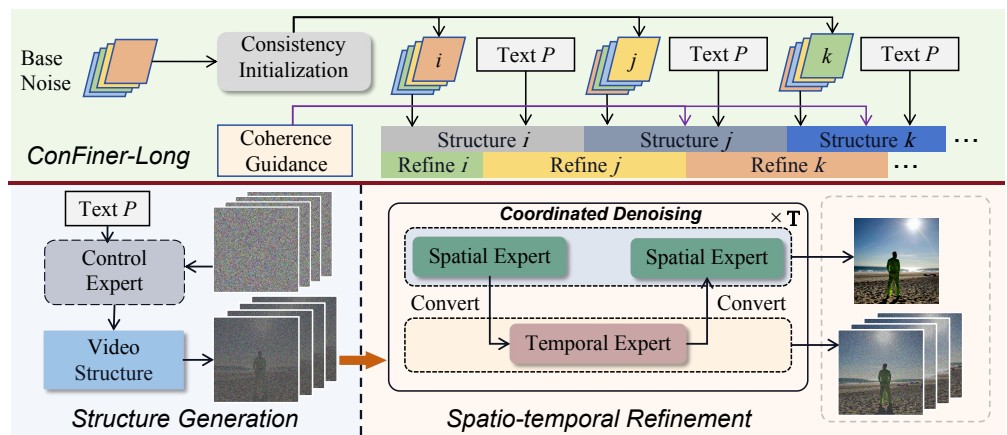

Figure 2: **Pipeline of Our ConFiner and ConFiner-Long.** ConFiner decouples the video generation process. Firstly, control expert generates a video structure. Subsequently, temporal and spatial experts perform the refinement of spatio-temporal details. Spatial and temporal experts work together with our coordinated denoising. By adding consistency initialization, coherence guidance and staggered refinement to ConFiner, ConFiner-Long can generate coherent long videos.

## 3.2 REVISITING DIFFUSION MODELS

The workflow of diffusion models consists of two processes: the forward process and the reverse denoising process. The forward process from timestep 0 to timestep $t$ can be expressed as follows:

$$\mathbf{x}_t = \sqrt{\overline{\alpha}_t}\mathbf{x}_0 + \sqrt{1 - \overline{\alpha}_t}\epsilon \tag{1}$$

where $\alpha_t = 1 - \beta_t$, $\overline{\alpha}_t = \prod_{i=1}^{t} \alpha_i$, $t$ is the diffusion step, $\epsilon$ is a random noise sampled from Standard Gaussian Distribution $\mathcal{N}(0, 1)$ and $\beta_t$ is a small positive constant between 0 and 1, representing the noise level of each timestep.

During the reverse denoising process, starting from a random noise at timestep $T$, the denoising model progressively predicts $\mathbf{x}_{t-1}$ from $\mathbf{x}_t$, ultimately getting the target data $\mathbf{x}_0$. Taking DDIM Song et al. (2020) as an example, the denoising model initially uses $\mathbf{x}_t$ to predict the noise. Then, $\mathbf{x}_t$ and the predicted noise are utilized together to predict $\mathbf{x}_0$ via the following expression:

$$\hat{\mathbf{x}}_0 = \frac{\mathbf{x}_t - \sqrt{1 - \overline{\alpha}_t}\epsilon_\theta^{(t)}(\mathbf{x}_t)}{\sqrt{\overline{\alpha}_t}} \tag{2}$$

where $\epsilon_\theta^{(t)}(x_t)$, $\hat{\mathbf{x}}_0$ represents the predicted noise and $\mathbf{x}_0$.

Then, based on the predicted noise and $\hat{\mathbf{x}}_0$, a prediction for $\mathbf{x}_{t-1}$ is derived as:

$$\mathbf{x}_{t-1} = \sqrt{\overline{\alpha}_{t-1}} \cdot \hat{\mathbf{x}}_0 + \sqrt{1 - \overline{\alpha}_{t-1}}\epsilon_\theta^{(t)}(\mathbf{x}_t) \tag{3}$$

By combining eq. (2) and eq. (3), single-step denoising can be expressed as:

$$\hat{\mathbf{x}}_0, \mathbf{x}_{t-1} = \text{Denoising}(\theta, \mathbf{x}_t, t, S) \tag{4}$$

where $S$ denotes the noise scheduler and $\theta$ represents the corresponding denoising model.

## 3.3 VIDEO STRUCTURE GENERATION

In the control stage, we select a video diffusion model skilled at handling video structure and employ it as control expert. The scheduler used in this expert can be denoted as $S_{\text{con}}$. During inference, to reduce computational overhead, we opt for a DDIM scheduler with a total inference step of $T_{i_1}$. When conducting inference, the list of timesteps utilized is: $[t_1(i_1), t_2(i_1), ..., t_{T_{i_1}}(i_1)]$. The selection of timesteps is made at uniform intervals.

---

**Algorithm 1** ConFiner (Control + Refinement)

---

1: **Input:** Prompt $P$, Control Expert $Con$, Spatial Expert $S$, Temporal Expert $T$, Noisy timestep $T_e$
2: **Output:** Generated video $\mathcal{V}$
3: // Control Stage
4: $\mathcal{V}_0 \leftarrow \text{Generate}(P, Con)$                 ▷ Generate video with coarse-grained details.
5: $Video\_Structure \leftarrow \text{Add noise}(\mathcal{V}_0, T_e, Con)$     ▷ Extract structure from low-quality video.
6: // Refinement Stage
7: $\mathcal{V}'_{T_e} \leftarrow Video\_Structure$
8: **for** each refinement step $T_k$ **do**
9:     **if** Standard Denoising **then**
10:       $\mathcal{V}'_{T_{k-1}} \leftarrow \text{Denoise}(\mathcal{V}'_{T_k}, T_k, S)$              ▷ Using spatial expert for denoising.
11:     **else if** Coordinated Denoising **then**
12:       $\mathcal{V}'_0(T_k) \leftarrow \text{Denoise}(\mathcal{V}'_{T_k}, T_k, S, P)$            ▷ Add spatial details.
13:       $\mathcal{V}''_{T_k} \leftarrow \text{Add noise}(\mathcal{V}'_0(T_k), T_k, T)$      ▷ Transfer to temporal expert's scheduler.
14:       $\mathcal{V}''_0(T_k) \leftarrow \text{Denoise}(\mathcal{V}''_{T_k}, T_k, T, P)$           ▷ Add temporal details.
15:       $\mathcal{V}'_{T_k} \leftarrow \text{Add noise}(\mathcal{V}''_0(T_k), T_k, S)$     ▷ Transfer to Spatial expert's scheduler.
16:       $\mathcal{V}'_{T_{k-1}} \leftarrow \text{Denoise}(\mathcal{V}'_{T_k}, T_k, S, P)$        ▷ Add spatial details and denoise.
17:     **end if**
18: **end for**
19: **Return** $\mathcal{V} = \mathcal{V}'_0$

---

After obtaining the timesteps list, we start with a random noise $\mathcal{V}_{t_{T_{i_1}(i_1)}}$ and progressively perform denoising over these timesteps, getting the first version of the video $\mathcal{V}_0$. Single-step sampling from eq. (4) can be rewritten as follows.

$$\hat{\mathcal{V}}_0(t_k(i_1)), \mathcal{V}_{t_{k-1}(i_1)} = \text{Denoising}(\theta_{\text{con}}, \mathcal{V}_{t_k(i_1)}, t_k(i_1), S_{\text{con}}) \tag{5}$$

where $\hat{\mathcal{V}}_0(t_k(i_1))$ represents the predicted $\mathcal{V}_0$ at timestep $t_k(i_1)$, $\mathcal{V}_{t_k(i_1)}$ denotes $\mathcal{V}$ at timestep $t_k(i_1)$, $\theta_{\text{con}}$ represents control expert and $S_{\text{con}}$ is the scheduler of control expert.

While we completed the entire sampling process to obtain the first version of video $\mathcal{V}_0$, the quality and coherence of the video are compromised due to our choice of a small $T_{i_1}$.

Therefore, we introduce $T_e$ steps of noise to $\mathcal{V}_0$. This operation is intended to create refinement opportunities for spatial and temporal experts. In this noise addition process, we utilize the Scheduler $S_s$ from the spatial expert used in refinement stage, resulting in the noisy video $\mathcal{V}'_{T_e}$ at timestep $T_e$. Transformed from eq. (1), this noise addition process can be expressed as:

$$\mathcal{V}'_{T_e} = \sqrt{\overline{\alpha}_{T_e}(S_s)} \cdot \mathcal{V}_0 + \sqrt{1 - \overline{\alpha}_{T_e}(S_s)} \cdot \epsilon \tag{6}$$

where $\overline{\alpha}_{T_e}(S_s)$ is the $\overline{\alpha}_t$ in scheduler $S_s$ at timestep $T_e$.

### 3.4 SPATIAL AND TEMPORAL DETAILS REFINEMENT

During the refinement stage, we add spatial and temporal details with spatial expert and temporal expert in the process of transforming $\mathcal{V}'_{T_e}$ to $\mathcal{V}'_0$. Similar to the control stage, we select $T_{i_2}$ steps for sampling between timestep $T_e$ and timestep 0. The list of timesteps used is: $[t_1(i_2), t_2(i_2), ..., t_{T_{i_2}}(i_2)]$.

Given that two experts respectively excel in spatial and temporal modeling, we aim to synergistically utilize both experts in the process of denoising $\mathcal{V}'_{T_e}$ to $\mathcal{V}'_0$, thus enhancing the spatio-temporal detail. A straightforward approach is alternating between the two experts at each timestep, leveraging the strengths of both models concurrently. In this case, eq. (4) can be rewritten as follows:

$$\hat{\mathcal{V}}_0(t_k(i_2)), \mathcal{V}'_{t_{k-1}(i_2)} = \text{Denoising}(\theta_X, \mathcal{V}'_{t_k(i_2)}, t_k(i_2), S_s)$$

$$\text{where } \theta_X = \begin{cases} \theta_S & \text{if } k \equiv 2 \pmod{0} \\ \theta_T & \text{if } k \equiv 2 \pmod{1} \end{cases} \tag{7}$$

where $\theta_S$, $\theta_T$ represent sptial expert and temporal expert, and $S_s$ denotes spatial expert's scheduler.

However, this method is ineffective because spatial expert and temporal expert are often on different noise scheduler. The data distributions for the spatial and temporal experts at the same timestep are inconsistent. The original data is on the scheduler of spatial expert, and directly switching to the scheduler of temporal expert at a certain timestep leads to conflicts and inconsistencies. To transform $\mathcal{V}'_{t_k(i_2)}$ to $\mathcal{V}'_{t_{k-1}(i_2)}$, we provide two options.

**Option 1 (Standard Denoising):** Since the original data $\mathcal{V}'_{T_e}$ is on the scheduler of spatial expert, we can directly employ the spatial expert for denoising at time step $t_k(i_2)$:

$$\hat{\mathcal{V}}_0(t_k(i_2)), \mathcal{V}'_{t_{k-1}(i_2)} = \text{Denoising}(\theta_S, \mathcal{V}'_{t_k(i_2)}, t_k(i_2), S_s) \tag{8}$$

**Option 2 (Coordinated Denoising):** Although two experts' schedulers differ, both schedulers share the same distribution at timestep 0. Hence, we can utilize timestep 0 to establish a connection between the two schedulers, facilitating the concurrent use of two experts within the same timestep. The specific details of this process are as follows.

First, at timestep $t_k(i_2)$, given $\mathcal{V}'_{t_k(i_2)}$, we employ the spatial expert for a one-step inference as eq. (8). After obtaining the predicted $\hat{\mathcal{V}}_0(t_k(i_2))$, it can be converted to $\mathcal{V}''_{t_k(i_2)}$ on the scheduler of temporal expert.

$$\mathcal{V}''_{t_k(i_2)} = \sqrt{\overline{\alpha}_{t_k(i_2)}(S_t)} \cdot \hat{\mathcal{V}}_0(t_k(i_2)) + \sqrt{1 - \overline{\alpha}_{t_k(i_2)}(S_t)} \cdot \epsilon \tag{9}$$

where $S_t$ represents the noise scheduler of temporal expert.

Then we can employ the temporal expert for denoising:

$$\hat{\mathcal{V}}_0(t_k(i_2)), \mathcal{V}''_{t_{k-1}(i_2)} = \text{Denoising}(\theta_T, \mathcal{V}''_{t_k(i_2)}, t_k(i_2), S_t) \tag{10}$$

This version of $\hat{\mathcal{V}}_0(t_k(i_2))$ predicted by temporal expert contains richer temporal information and demonstrates enhanced inter-frame coherence. Subsequently, we transform $\hat{\mathcal{V}}_0(t_k(i_2))$ using the scheduler of spatial expert into a $\mathcal{V}'_{t_k(i_2)}$ with more extensive temporal information.

$$\mathcal{V}'_{t_k(i_2)} = \sqrt{\overline{\alpha}_{t_k(i_2)}(S_s)} \cdot \hat{\mathcal{V}}_0(t_k(i_2)) + \sqrt{1 - \overline{\alpha}_{t_k(i_2)}(S_s)} \cdot \epsilon \tag{11}$$

Finally, the spatial expert is used again to predict $\mathcal{V}'_{t_{k-1}(i_2)}$ including augmented spatio-temporal information as eq. (8).

### 3.5 CONFINER-LONG FRAMEWORK

We also leverage ConFiner to design a pipeline for long video generation. This pipeline generate multiple short video segments and introduces three strategies to ensure consistency and coherence between these segments.

First, we design consistency initialization strategy to promote consistency between segments. The initial noise affects the content of the generated video significantly. To improve the consistency between segments, we first sample a $Noise\_base \in \mathbf{R}^{H \times W \times C \times F}$, which is then subjected to frame-wise shuffling to obtain the initial noise for each segment. Sharing base noise enhances the content consistency between segments while shuffling maintains a little randomness.

Additionally, we introduce a staggered refinement mechanism to further improve the overall coherence of the video. In our segmented generation approach, the transition points between segments tend to exhibit the highest inconsistency. Therefore, in long video generation, we perform the Control Stage and Refinement Stage in a staggered manner. Specifically, the latter half of the preceding structure and the former half of the succeeding structure are used as inputs for a same refinement pass. The refinement stage can seamlessly stitch the two structures together, which ensures a more natural and smoother transition between segments.

Table 1: **Objective Evaluation Results.** In this experiment, ConFiner utilized AnimateDiff-Lightning as the control expert and selected stable diffusion 1.5 for spatial expert. Lavie and Modelscope T2V are chosen as temporal expert.

| Method | Inference Steps | Subject Consistency↑ | Motion Smoothness↑ | Aesthetic Quality ↑ | Imaging Quality ↑ |
|---|---|---|---|---|---|
| Lavie Wang et al. (2023c) | 10 | 0.940 | 0.967 | 0.570 | 0.658 |
| Lavie Wang et al. (2023c) | 20 | 0.954 | 0.966 | 0.587 | 0.683 |
| Lavie Wang et al. (2023c) | 50 | 0.958 | 0.965 | 0.597 | 0.696 |
| Lavie Wang et al. (2023c) | 100 | 0.957 | 0.965 | 0.596 | 0.695 |
| AnimateDiff-Lightning Lin & Yang (2024) | 10 | 0.983 | 0.983 | 0.635 | 0.689 |
| AnimateDiff-Lightning Lin & Yang (2024) | 20 | 0.984 | 0.980 | 0.636 | 0.697 |
| AnimateDiff-Lightning Lin & Yang (2024) | 50 | 0.981 | 0.971 | 0.638 | 0.705 |
| AnimateDiff-Lightning Lin & Yang (2024) | 100 | 0.977 | 0.964 | 0.623 | 0.699 |
| Modelscope T2V Wang et al. (2023b) | 10 | 0.983 | 0.980 | 0.570 | 0.670 |
| Modelscope T2V Wang et al. (2023b) | 20 | 0.985 | 0.980 | 0.575 | 0.702 |
| Modelscope T2V Wang et al. (2023b) | 50 | 0.988 | 0.990 | 0.592 | 0.716 |
| Modelscope T2V Wang et al. (2023b) | 100 | 0.987 | 0.990 | 0.594 | 0.715 |
| **ConFiner** w/ Lavie | 9 | 0.993 | 0.991 | 0.699 | 0.734 |
| **ConFiner** w/ Lavie | 18 | 0.993 | 0.990 | 0.703 | 0.739 |
| **ConFiner** w/ Modelscope | 9 | 0.994 | 0.991 | 0.698 | 0.731 |
| **ConFiner** w/ Modelscope | 18 | **0.994** | **0.991** | **0.707** | **0.739** |

Although consistency initialization and staggered refinement have ensured content consistency and smooth transitions between segments, if the motion modes of video structures are not coherent, it will be impossible to combine them into a reasonable long video. Thus, we propose a coherent guidance to promote the motion mode of new segment to follow the preceding segment. In video generation, predicted noises affect the direction of generation and determine the motion mode. So we generate each structure one by one, using noises of the previous segments to guide the subsequent structure. Specifically, during the sampling process, we use the gradient of the L2 loss to guide the sampling direction. The L2 loss is calculated between the predicted noise of the current segment and the noise in the previous segment. The guided noise is calculated as follows:

$$\epsilon_t^{S_2} = \hat{\epsilon}_t^{S_2} - \gamma \nabla_{\hat{\epsilon}_t^{S_2}} \|\hat{\epsilon}_t^{S_2} - \epsilon_t^{S_1}\|^2 \tag{12}$$

where $\hat{\epsilon}_t^{S_2}$ represents the noise of current segment predicted by denoising model at timestep $t$, $\epsilon_t^{S_1}$ is the noise of former segment at timestep $t$ and $\gamma$ is a constant.

In this way, coherent guidance can make the noise of the two segments similar, which allows the motion mode of the latter segment to inherit that of the previous segment. Additionally, coherence guidance also reduces the pixel distance between noises of two segments, which can help maintain content consistency between segments.

## 4 EXPERIMENTS

In the experiment, we selected AnimateDiff-Lightning Lin & Yang (2024) as control expert, and Stable Diffusion 1.5 Rombach et al. (2022) as the spatial expert. For the temporal expert, we opted for two open-source models, lavie Wang et al. (2023c) and modelscope Wang et al. (2023b).

### 4.1 OBJECTIVE EVALUATION

For objective evaluation experiments, we utilized the cutting-edge benchmark, Vbench Huang et al. (2023). Vbench provides 800 prompts that test various capabilities of video generation models. In our experiments, each model generated 800 videos using these prompts, and the resulting videos were assessed using four metrics to evaluate their Temporal Quality and Frame-wise Quality.

Table 2: **Subjective Evaluation Results.** Each model generates videos using the top 100 prompts from Vbench Huang et al. (2023). The videos were evaluated by 30 users, with each video being rated as good, normal, or bad on three dimensions.

| Method | Coherence | | | Text-Match | | | Visual Quality | | |
|---|---|---|---|---|---|---|---|---|---|
| | Bad↓ | Normal~ | Good↑ | Bad↓ | Normal~ | Good↑ | Bad↓ | Normal~ | Good↑ |
| AnimateDiff-Lightning | 0.37 | 0.42 | 0.21 | **0.06** | 0.51 | 0.43 | 0.29 | 0.51 | 0.20 |
| Modelscope T2V | 0.14 | 0.48 | 0.38 | 0.21 | 0.53 | 0.26 | 0.34 | 0.45 | 0.21 |
| Lavie | 0.11 | 0.46 | 0.43 | 0.24 | 0.46 | 0.30 | 0.32 | 0.49 | 0.19 |
| **ConFiner** w/ Lavie | 0.08 | 0.43 | 0.49 | 0.08 | 0.48 | **0.44** | 0.13 | 0.36 | **0.51** |
| **ConFiner** w/ Modelscope | **0.07** | 0.42 | **0.51** | 0.08 | 0.50 | 0.42 | **0.09** | 0.41 | 0.50 |

Table 3: **Ablation Study of** $T_e$. In most cases, as $T_e$ increases, the temporal metric decreases and the imaging quality improves. However, when the control stage involves only 4 steps, too high values of $T_e$ (such as 300 or 500) can lead to imaging collapse.

| Method | Control Stage Steps | $T_e$ | Subject Consistency↑ | Motion Smoothness↑ | Aesthetic Quality ↑ | Imaging Quality↑ |
|---|---|---|---|---|---|---|
| **ConFiner** w/ Lavie | 4 | 50 | **0.993** | **0.991** | 0.703 | 0.733 |
| **ConFiner** w/ Lavie | 4 | 100 | **0.993** | 0.990 | 0.702 | 0.737 |
| **ConFiner** w/ Lavie | 4 | 200 | 0.992 | 0.989 | **0.710** | **0.744** |
| **ConFiner** w/ Lavie | 4 | 300 | 0.978 | 0.986 | 0.383 | 0.303 |
| **ConFiner** w/ Lavie | 4 | 500 | 0.967 | 0.983 | 0.338 | 0.265 |
| **ConFiner** w/ Modelscope | 4 | 50 | **0.995** | **0.991** | 0.701 | 0.733 |
| **ConFiner** w/ Modelscope | 4 | 100 | 0.994 | **0.991** | 0.698 | 0.733 |
| **ConFiner** w/ Modelscope | 4 | 200 | 0.994 | 0.990 | **0.712** | **0.736** |
| **ConFiner** w/ Modelscope | 4 | 300 | 0.990 | 0.987 | 0.560 | 0.429 |
| **ConFiner** w/ Modelscope | 4 | 500 | 0.993 | 0.992 | 0.513 | 0.370 |
| **ConFiner** w/ Lavie | 8 | 50 | **0.994** | **0.991** | 0.708 | 0.741 |
| **ConFiner** w/ Lavie | 8 | 100 | 0.993 | 0.990 | 0.706 | 0.739 |
| **ConFiner** w/ Lavie | 8 | 200 | 0.991 | 0.989 | 0.716 | 0.742 |
| **ConFiner** w/ Lavie | 8 | 300 | 0.983 | 0.985 | 0.718 | 0.744 |
| **ConFiner** w/ Lavie | 8 | 500 | 0.978 | 0.980 | **0.721** | **0.751** |
| **ConFiner** w/ Modelscope | 8 | 50 | **0.994** | **0.991** | 0.708 | 0.740 |
| **ConFiner** w/ Modelscope | 8 | 100 | 0.994 | **0.991** | 0.707 | 0.739 |
| **ConFiner** w/ Modelscope | 8 | 200 | 0.993 | 0.990 | 0.716 | 0.742 |
| **ConFiner** w/ Modelscope | 8 | 300 | 0.992 | 0.989 | 0.720 | 0.747 |
| **ConFiner** w/ Modelscope | 8 | 500 | 0.991 | 0.987 | **0.727** | **0.752** |

For Temporal Quality Metrics, we use Subject Consistency and Motion Smoothness. For Framewise Quality Metrics, we use Aesthetic Quality and Imaging Quality.

In this experiment, we employed AnimateDiff-Lightning, Lavie, and mocelscope T2V to generate over total timesteps of 10, 20, 50, and 100. We then utilize our ConFiner to conduct generation with 9(4+5) and 18(8+10) timesteps, where $T_e$ is set to 100. All evaluation results are presented in table 1. Each individual experiment can be completed in 3-5 hours on a single RTX 4090. In each experiment, we repeated for five times with different random seeds.

## 4.2 SUBJECTIVE EVALUATION

In our subjective evaluation, we employed our ConFiner with 18 inference steps to generate videos using the top 100 prompts from Vbench. These videos were evaluated alongside those generated by AnimateDiff-Lightning, Modelscope T2V, and Lavie with 50-step inference, by 30 users. Users rated each video across three dimensions: coherence, text-match, and visual quality, each dimension being categorized into three levels: good, normal, and bad. The scoring results are shown in table 2.

Table 4: Comparison of Training and Inference Time.

| Time Cost | ConFiner | LavieWang et al. (2023c) | Animate DiffusionGuo et al. (2023) | ModelscopeWang et al. (2023b) |
|---|---|---|---|---|
| Training | 0 | $> 100\times$ *A100 day* | $> 100\times$ *A100 day* | $> 100\times$ *A100 day* |
| Inference | $\approx 5S$ | *>1min* | *>1min* | *>1min* |

Table 5: **Ablation Study of Denoising Type.**

| Method | Inference Steps | Denoising Type | Subject Consistency↑ | Motion Smoothness↑ | Aesthetic Quality ↑ | Imaging Quality ↑ |
|---|---|---|---|---|---|---|
| **ConFiner** w/ Lavie | 9 | **Coordinated Denoising** | 0.993 | 0.991 | 0.699 | 0.734 |
| **ConFiner** w/ Lavie | 9 | Only Temporal Expert | 0.994 | 0.993 | 0.552 | 0.618 |
| **ConFiner** w/ Lavie | 9 | Only Spatial Expert | 0.883 | 0.907 | 0.749 | 0.766 |
| **ConFiner** w/ Lavie | 18 | **Coordinated Denoising** | 0.993 | 0.990 | 0.703 | 0.739 |
| **ConFiner** w/ Lavie | 18 | Only Temporal Expert | 0.993 | 0.991 | 0.583 | 0.632 |
| **ConFiner** w/ Lavie | 18 | Only Spatial Expert | 0.859 | 0.880 | 0.754 | 0.758 |
| **ConFiner** w/ Modelscope | 9 | **Coordinated Denoising** | 0.994 | 0.991 | 0.698 | 0.731 |
| **ConFiner** w/ Modelscope | 9 | Only Temporal Expert | 0.995 | 0.993 | 0.518 | 0.599 |
| **ConFiner** w/ Modelscope | 9 | Only Spatial Expert | 0.912 | 0.922 | 0.732 | 0.758 |
| **ConFiner** w/ Modelscope | 18 | **Coordinated Denoising** | 0.994 | 0.991 | 0.707 | 0.739 |
| **ConFiner** w/ Modelscope | 18 | Only Temporal Expert | 0.993 | 0.992 | 0.577 | 0.641 |
| **ConFiner** w/ Modelscope | 18 | Only Spatial Expert | 0.861 | 0.893 | 0.765 | 0.772 |

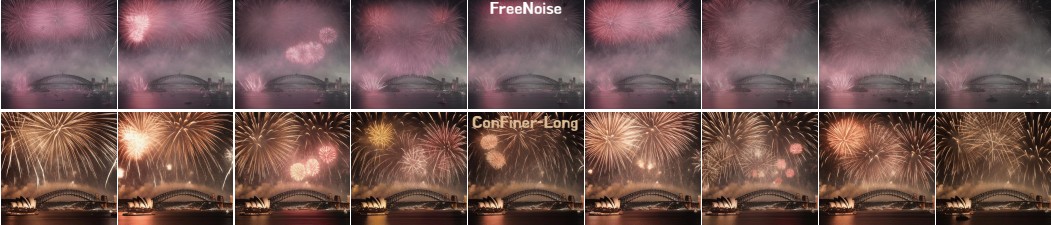

"A spectacular fireworks display over Sydney Harbour, 4K, high resolution"

Figure 3: Comparison of Our ConFiner-Long with FreeNoise Qiu et al. (2023b).

### 4.3 COMPARISON OF COMPUTATION EFFICIENCY

In this section, we compare the training and inference cost of our ConFiner with other video diffusion models. The results are shown in table 4.

### 4.4 ABLATION STUDY ON CONTROL AND REFINEMENT LEVEL

As eq. (6), we apply noise for $T_e$ steps to the videos generated during the control stage to create optimization space for the refinement stage. A larger $T_e$ value increases the impact of the refinement stage. For the four settings same as objective experiment, we set $T_e$ to 50, 100, 200, 300, and 500, with other experimental settings consistent. The performance comparison is shown in table 3.

### 4.5 ABLATION STUDY ON COORDINATED DENOISING

To verify the effectiveness of coordinated denoising, we conducted ablation experiments on the denoising type during the refinement stage. Specifically, in this experiment, we used Lavie and ModelScope as the temporal experts, setting the total inference steps to 9 and 18, respectively, thus constructing four experimental settings. For each setting, we refined using three different denoising types during the refinement stage: using coordinated denoising; using only the temporal expert; and using only the spatial expert. The performance of the three denoising types is shown in table 5.

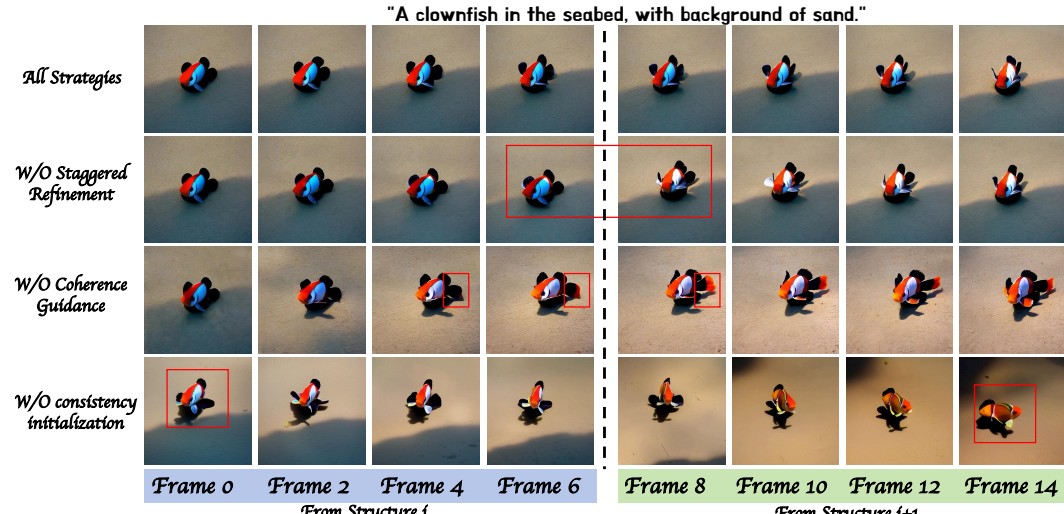

Figure 4: Ablation Study on Three Strategies of ConFiner-Long.

## 4.6 ABLATION STUDY ON STRATEGIES OF CONFINER-LONG

In this section, we conducted ablation experiments on three strategies within the ConFiner-Long framework. Using the same preceding video segments, we generated subsequent video segments with either all strategies or only two. The visual comparison of the four video segments against the preceding one is shown in fig. 4. The overall visual comparison between ConFiner-Long and the existing training-free long video generation method FreenoiseQiu et al. (2023b) is shown in fig. 3.

## 5 CONCLUSION

In this paper, we introduce ConFiner, a training-free framework that can generate high-quality videos with chain of diffusion experts. It decomposes video generation into three components: structure control, spatial refinement and temporal refinement. Each component is handled by a off-the-shelf expert that specializes in this task. Additionally, we propose coordinated denoising to enable two expert cooperate when denoising. We also design ConFiner-Long framework to generate long coherent videos. Experimental results confirm that our ConFiner enhances the aesthetics and coherence of generated videos while reducing sampling time significantly. And our ConFiner-Long can generate consistent and coherent videos with up to 600 frames. Our approach paves the way for cost-effective new possibilities in filmmaking, animation production, and video editing.

### ETHICS STATEMENT

While ConFiner has significant potential for creative industries, such as filmmaking and animation, we recognize the potential misuse of such technology for generating misleading or harmful content. To mitigate these risks, we advocate for responsible use in line with industry standards and ethical guidelines, and will include an NSFW detector in the open-source code.

### REPRODUCIBILITY STATEMENT

We utilize an open-source evaluation framework Vbench, and our experiments are conducted using prompts publicly available from Vbench, ensuring the reproducibility of our results. The code for our ConFiner framework, including the implementation details, is submitted in the appendix. Moreover, we have provided a comprehensive README file along with a detailed guide for reproducing the experimental metrics.

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
