# OpenReview forum: "Training-free Long Video Generation with Chain of Diffusion Model Experts"
_ICLR.cc/2025/Conference — ICLR 2025 Conference Withdrawn Submission_

### Official Review · Reviewer_Py7V · 2024-10-27

**Soundness:** 3
**Presentation:** 4
**Contribution:** 3
**Rating:** 5
**Confidence:** 5

**Summary:**

This paper introduces ConFiner, a method that uses image diffusion models to improve video generation. ConFiner first uses a text-to-video model to generate a rough structure of the video. Then, noise is added to this video, and it's passed through image generation models for spatial refinement and the video model for temporal refinement. They also propose ConFiner-Long, which is designed for making long videos by using three strategies to keep segments consistent and transitions smooth. Experimental results show that ConFiner improves video quality, and ConFiner-Long successfully generates longer videos.

**Strengths:**

a. Clarity and Simplicity: The approach presented is straightforward, and the method section is generally clear and easy to follow.
b. Comprehensive and Convincing Experiments: The experiments are thorough, with results showing that ConFiner effectively improves video quality and coherence compared to existing models.
c. Long Video Generation Capability: ConFiner-Long introduces three strategies that help maintain consistency and smooth transitions in longer videos, allowing for the generation of high-quality videos with extended frame lengths.

**Weaknesses:**

1. The title suggests a focus on “training-free long video generation”, but the main content is mainly about introducing ConFiner’s ability to enhance video quality. And most experiments also focus on ConFiner’s improvements, creating a bit of a disconnect between the title and the paper's main content.
2. Limited Novelty: ConFiner’s approach of using T2I models to enhance T2V quality isn’t new. Similar ideas have already been explored in works like VideoElevator[1]. This reduces the novelty of the proposed method.
3. Missing Related Work: The paper is notably lacking in its discussion of long video generation and related work using T2I as a video generation refiner, such as [1,2,3] This aspect is vital as it forms the basis of the research's motivation. The omission of these most related studies is puzzling.
4. The experiments mainly focus on ConFiner's comparison and analysis and lacks comparison with existing long video generation methods, like StoryDiffusion, StreamingT2V, SEINE.
5. The ablation study on three strategies of ConFiner-Long is missing quantitative results. Fig.4 cannot fully prove the effectiveness of three strategies.


[1] VideoElevator: Elevating Video Generation Quality with Versatile Text-to-Image Diffusion Models
[2] StoryDiffusion: Consistent Self-Attention for Long-Range Image and Video Generation
[3] SEINE: Short-to-Long Video Diffusion Model for Generative Transition and Prediction

**Questions:**

1. The writing of the paper needs to correspond to the training-free long video generation in the title, including motivation, related work, method, and experiments.
2. In the related work, the omission of these most related studies is puzzling.
3. The authors should explain how their method is different from current methods and what makes it stand out, including ConFiner and ConFiner-Long

---

### Official Review · Reviewer_Fq3C · 2024-10-31

**Soundness:** 3
**Presentation:** 3
**Contribution:** 3
**Rating:** 5
**Confidence:** 5

**Summary:**

This paper proposes ConFiner that decouples the video generation task into three sub-tasks. For each sub-task, ConFiner uses three off-the-shelf diffusion models as experts. Furthermore, ConFiner proposes coordinated denoising, which can allow different diffusion models collaborate timestep-wise in video generation process.

**Strengths:**

1. This paper discusses how to generate high-quality videos with already trained models, which is a very interesting topic.
2. The structure of this paper is well-organized and easy to follow.
3. The experimental results show the effectiveness of the proposed method.

**Weaknesses:**

There are some questions.
1. On line 283, the author claims that "both schedulers share the same distribution at timestep 0." However, the distribution at timestep 0 corresponds to that of the training dataset. Typically, the training datasets used for T2V (Text-to-Video) and T2I (Text-to-Image) are not identical, so this statement is somewhat inaccurate. I suggest the authors provide additional insights into the choice of using timestep 0 as the anchor for the generated image or video.
2. The authors add a certain amount of noise to the video or image generated at each stage when using each expert. I am curious whether the final generated video retains any connection to the original structured video.
3. In the section on the consistency initialization strategy (from line 313 to line 317), does the author use the same initial noise for each short video clip, with only the frame order randomized in each initial noise? If so, would this lead to repetitive content in the subsequently generated videos?
4. From lines 348 to 361, the authors use L2 loss to calculate the difference between the current noise and the previous segment noise. However, according to the consistency initialization strategy, the noise is predefined. This raises some confusion—why is it necessary to further optimize the noise input to the model?
5. In the video demo, I observed that ConFiner generates smoother videos. Additionally, compared to the base T2V model, the colors in ConFiner’s videos tend to appear more grayish.

**Questions:**

Please see above. If the author solves my problems, I will consider raising the score. Thanks.

---

### Official Review · Reviewer_s9VB · 2024-11-03

**Soundness:** 2
**Presentation:** 3
**Contribution:** 1
**Rating:** 3
**Confidence:** 4

**Summary:**

The paper proposes a framework for long video generation by ensembling multiple diffusion models. Video generation is decomposed into spatial and temporal parts. T2V models are employed for control experts and temporal experts, and T2I models are employed for spatial experts. The authors utilize the timestep 0 as the connection to better ensemble different diffusion model experts. The proposed method can generate more consistent video frames.

**Strengths:**

1. The paper is well-written and easy to follow. The authors conduct extensive experiments to support the claims in the paper.
2. The paper proposes ConFiner by decomposing video generation into three subtasks. Multiple expert diffusion models are employed.
3. Coordinated denoising is proposed to allow two experts on different noise schedulers to collaborate timestep-wise in the video generation process.
4. The proposed method supports longer video generation.

**Weaknesses:**

1. The proposed method can generate longer videos with more frames. However, only the number of frames increases, neither the content nor the dynamics of generated videos increases. From Fig. 1, the motions of StreamT2V are larger. Also from examples on the project page, the motion of the proposed method is small. Therefore, the video itself is not long indeed.
2. The most significant contribution of this paper is the coordinated denoising. It is to use the timestep 0 as the connection, which requires denoising the latent to timestep 0 in the intermediate steps. It increases the computation costs. Furthermore, this technique is more like a trick.
3. In the experiment part, the comparison methods are not state-of-the-art. The author should compare with more state-of-the-art methods.
4. The technical contributions of the paper are not significant enough.

**Questions:**

Please see my concerns in the weakness part.

---

### Official Review · Reviewer_ZQHU · 2024-11-03

**Soundness:** 2
**Presentation:** 2
**Contribution:** 2
**Rating:** 5
**Confidence:** 5

**Summary:**

This paper introduces ConFiner, a model that decouples the video generation task into distinct sub-tasks: structure control, spatial-temporal refinement. It employs three pre-existing diffusion experts, each responsible for a specific task, thereby reducing the overall burden on the model and enhancing both the quality and speed of video generation. The paper further proposes a method of coordinated denoising, enabling two experts with different noise schedulers to collaborate on a timestep basis during the video generation process. Expanding on the ConFiner framework, the paper outlines three strategies: a consistency initialization strategy, a staggered refinement mechanism, and coherent guidance, which together aim to construct ConFiner-long, a model designed to generate long videos.

**Strengths:**

1. The paper proposes a novel framework that utilizes both ready-made text-to-video and text-to-image models to perform video generation.
2. The experimental results show that ConFiner can generate higher quality videos on both objective and subjective metrics with a 10% reduction in costs.
3. ConFiner-Long can generate consistent and coherent videos up to 600 frames.

**Weaknesses:**

1. The generated videos exhibit low dynamics. It seems that ensuring video consistency is quite conflicting with achieving dynamics.
2. The contribution may be considered weak, as it heavily relies on other works, and some current video generations have presented much better video generation capabilities.
3. The core idea of splitting video generation into three stages is reasonable, but there lacks more analysis on why it must be split into three stages specifically.

**Questions:**

1. Provide some results that exhibit better dynamics.
2. How does Confiner process videos of different resolutions if the models used are trained with different resolutions?
3. If I want to generate some special videos, but some of the text-to-video models lack the ability to generate such kinds of videos, how can this be resolved? For example, I want to use a LoRA checkpoint with some special cartoon character for the text-to-image model. The other text-to-video model can generate such character structures or motions. How can this be resolved?

---

### Note · Authors · 2024-11-13

I have read and agree with the venue's withdrawal policy on behalf of myself and my co-authors.